# Research and Application of MPPT Control Strategy Based on Improved Slime Mold Algorithm in Shaded Conditions

**Changxin Fu** [1], **Lixin Zhang** [1,2,*] **and Wancheng Dong** [1]

1    College of Mechanical and Electrical Engineering, Shihezi University, Shihezi 832000, China;
fuchangxin@shzu.edu.cn (C.F.); dongwancheng@shzu.edu.cn (W.D.)
2    Industrial Technology Research Institute, Shihezi 832003, China
*    Correspondence: zhanglixin@shzu.edu.cn

**Abstract:** A PV maximum power tracking strategy for shaded conditions, based on an improved slime mold algorithm, is proposed in this research. To verify the superiority of the proposed algorithm, four bionomics algorithms—particle swarm optimization (PSO), tuna swarm optimization (TSO), squirrel search algorithm (SSA), and black widow spider algorithm (BWO)—were compared. The output parameter of the five control algorithms was summarized and analyzed. The adaptability of the algorithms was proven by setting different shading conditions. The simulation results demonstrated that the proposed algorithm possessed short response time, good tracking effect and fewer fluctuations. Eventually, the different algorithms were verified in the HIL + RCP physical platform. The experimental outcomes showed that the improved slime mold algorithm possessed the best tracking effect, with fewer power fluctuations.

**Keywords:** under shading conditions; MPPT; improved slime mold algorithm; HIL + RCP

## 1. Introduction

With the depletion of fossil energy, renewable energy sources are gaining significant attention. Considering the resources' characteristics, such as being inexhaustible, sustainable, and zero-pollution [1,2], photovoltaic power generation is widely applied, worldwide [3]. It has a broad range of applications, from small solar appliances to photovoltaic power plants of several hundred megawatts [4,5]. However, photovoltaic power generation depends on outside temperature and radiance [6]. In particular, when the surface layer of PV cells is covered by shadow, the efficiency of output power drops significantly [7,8]. As such, the P–U characteristic curve of PV power generation is converted into a multi-peak curve from the single-peak curve, under shading conditions [9,10]. Tracking the global maximum power point (GMPP) of the shaded PV process ensures the output power, and also guarantees the PV modules to operate properly [11–13].

Conventional maximum power tracking methods (perturbation and interference) [14], incremental conductance (INC), and variable step improvement algorithm) are effective when light is uniform. However, they tend to fall into the local maximum power point (LMPP) under shaded conditions [15]. The unreasonable output power can lead to residual energy inside the PV cell, causing a "hot spot" effect [16,17]. Therefore, traditional algorithms may no longer be suitable for tracking shaded photovoltaic power.

Based on this, relevant scholars have researched a variety of GMPP control algorithms, which mainly consist of the adaptive control algorithm and the bionic optimization algorithm. The algorithm can be summarized as follows. Through the controller's structural improvement, such as PID control [18], fuzzy logic control (FLC) [19,20], sliding mode control (SMC) [21], etc., a feedback-based closed-loop control process can be achieved. However, the output of the parameters may be poor. Therefore, more and more bionic algorithms have introduced shaded PV, whose characteristics are summarized as follows: information sharing among individuals, and unique position update methods. Bionic

algorithms include genetic algorithms (GA) [22] and differential evolution (DE) [23,24]. Population optimization algorithms include particle swarm algorithm (PSO) [25,26], gray wolf swarm algorithm (GWO) [27], cuckoo search algorithm (CS) [28,29], whale swarm optimization algorithm (WOA) [24,30], bat optimization algorithm (BA) [31,32], ant colony algorithm (ACO) [33,34], and squirrel search algorithm (SSA) [35], etc. Compared with traditional algorithms, MPPT based on the bionic algorithm has higher tracking efficiency, a shorter response time, and better flexibility and adaptability. Each algorithm has varying position update mechanisms and iterative approaches. Some have a position updating process obtained by generating random numbers (PSO, GWO, etc.). CS applies a probabilistic switching parameter to relocate the position randomly via Levy flight. It is crucial to find the appropriate bionic algorithm.

Slime mold algorithm (SMA) is an optimization algorithm, proposed in 2020, based on the foraging behavior of slime molds [36]. In the process of finding food, the slime mold possesses the property of contracting. Meanwhile, according to the quality of food source, a network of veins of varying thicknesses will be formed. Additionally, when acquiring food sources, the slime mold has a certain probability to search in unknown regions. Therefore, the algorithm is widely applied in power systems. A solution for the distribution network reconfiguration (DNR) problem with distributed generation (DG) based on the parallel slime mold algorithm (PSMA) in [37] was used to solve the DNR problem with DG more accurately and quickly. The authors of [38] proposed an advanced slime mold algorithm (SMA) integrated Nelder–Mead simplex strategy and chaotic map, called CNMSMA. Chaotic maps replace the random number rand that affects the choice of location updating strategy to improve exploratory patterns. The authors of [39] proposed efficient and robust MPPT controllers using novel slime mold optimization (SMO) and improved salp swarm optimization algorithm (ISSA) to track GMPP for different PV array configurations. However, the above algorithms only had a single method of updating position. Response time and stability cannot be guaranteed during multi-peak curve maximum point tracking. Therefore, a novel slime mold optimization algorithm is proposed in this research. Two aspects of the algorithm are optimized. First, a nonlinear decreasing strategy is proposed to improve the global search time by modifying the formula of parameter *a*. Secondly, a spiral search strategy is added, which expands the algorithm's search range and improves the global search performance.

To verify the superiority of the improved slime mold algorithm (ISMA), the control model was constructed in Matlab/Simulink platform. Particle swarm algorithm (PSO), squirrel search algorithm (SSA), tuna swarm algorithm (TSO), and black widow spider algorithm (BWO) were introduced for comparison in the following section. The experimental validation of the proposed algorithm was performed through the HIL + RCP platform, and the consequences were analyzed. Conclusions were drawn in the final section.

## 2. Partial Shading and Photovoltaic Systems

### 2.1. Photovoltaic Model and Characteristics

Photovoltaic shadows are mainly caused by surface dust, clouds, and buildings, etc. The output characteristics are extremely susceptible to non-linearity and the external environment. Under shading conditions, the characteristic curve may be converted from a single-peak curve to a multi-peak curve. As the situation intensifies, the process of GMPP tracking becomes complicated. The final tracking power may fall into LMPP, and, the efficiency of PV generation may even decrease by 70%. On the other hand, this situation can easily cause a "hot spot" effect, which can damage PV cells. Figure 1 demonstrates the schematic of shaded PV.

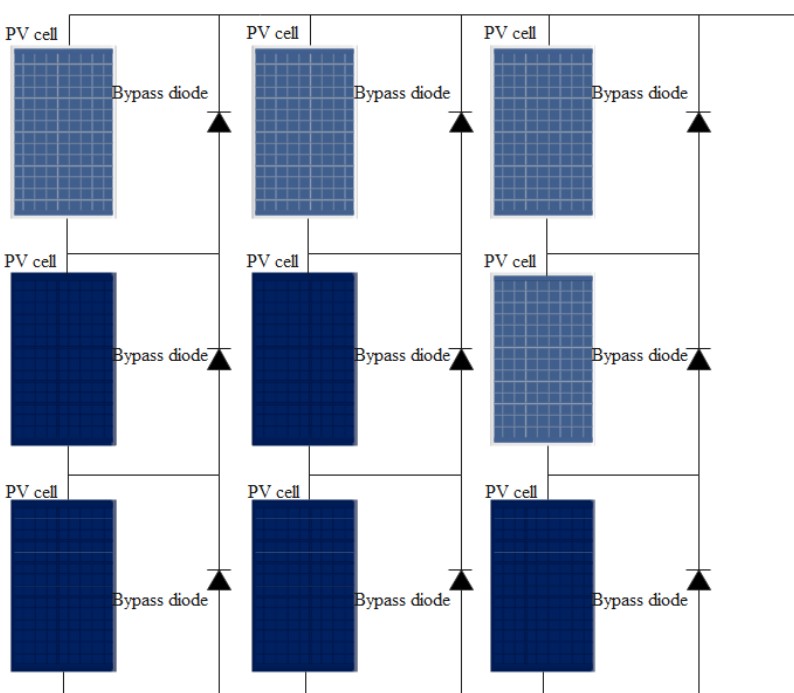

**Figure 1.** Schematic diagram of the presence of shadows during photovoltaic power generation.

The output characteristic curve of the PV power system under uniform light conditions is demonstrated in Figure 2. Figure 3 illustrates the PV power output characteristic curve under shading conditions.

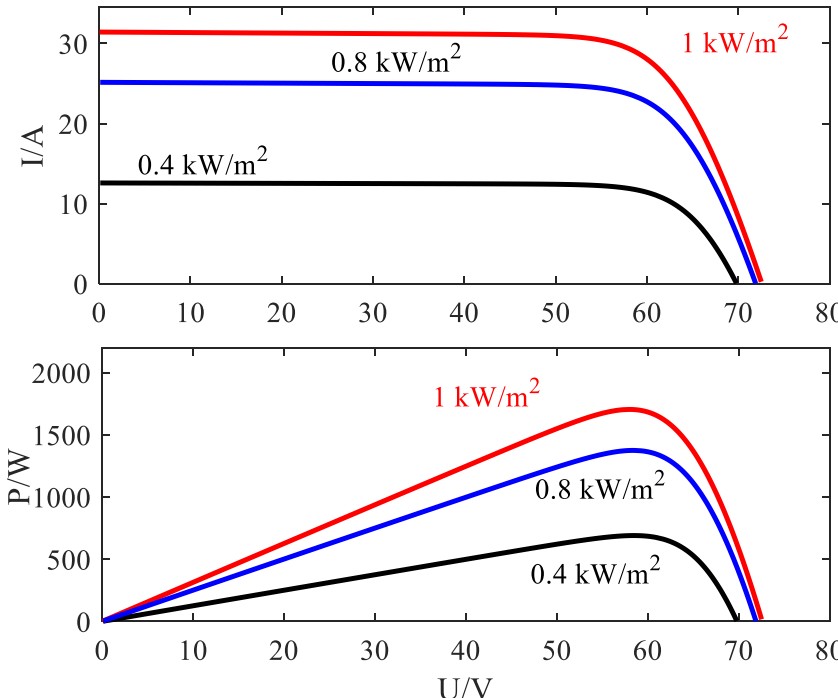

**Figure 2.** Output characteristics of photovoltaic cells under uniform radiance.

Radiance and temperature are the two main factors of the PV system [11,40]. Photovoltaic cells are equated as power supply, and the circuit model also contains diode, resistance and load. The whole circuit is displayed in schematic form in Figure 4.

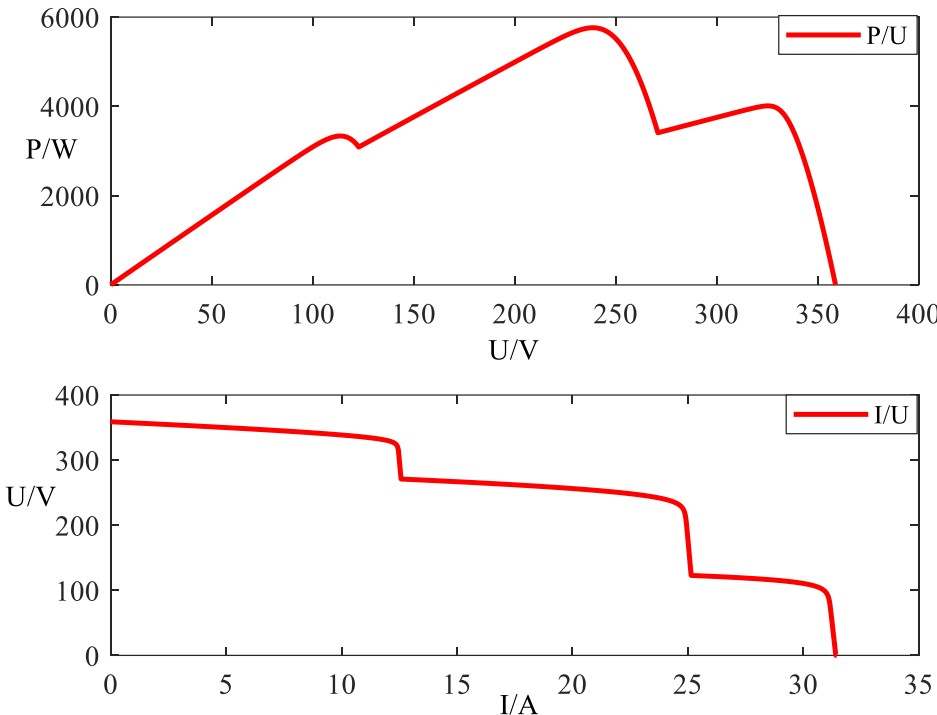

**Figure 3.** Output characteristics curve of PV power generation under uneven radiance.

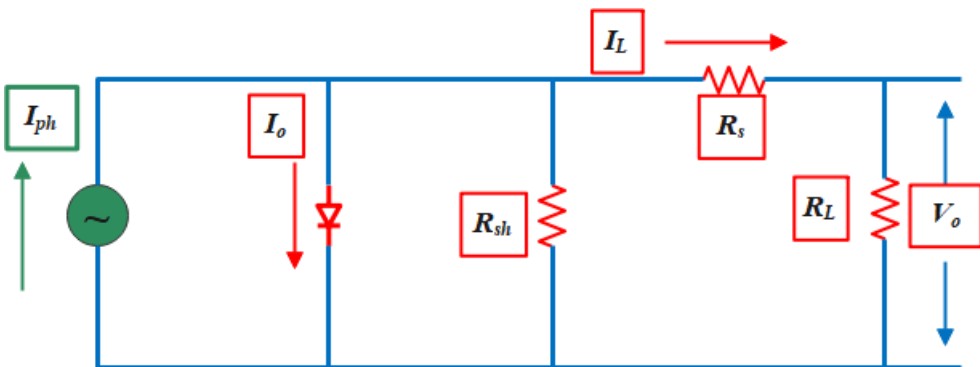

**Figure 4.** Mathematical model of photovoltaic cells.

According to the above circuit, the mathematical model of the PV cell can be deduced as:

$$I = I_{ph} - I_o \ \exp^{\frac{q(V_{pv}+R_sI)}{AKT}} - \frac{V_{pv} + R_s I}{R_{sh}} \tag{1}$$

where $I_{ph}$ is the output current of PV, $V_o$ is the output voltage of PV, $R_s$ denotes the equivalent series resistance, $T$ is the PV module temperature, $K$ is the Boltzman constant, $A$ is the ideality factor, $I_L$ is the shunt current, $R_{sh}$ is the equivalent shunt resistance, and $I_L$ denotes the current of $R_L$, and $R_L$ is the resistance of load.

### 2.2. System Description

The process of MPPT can be summarized as follows. The voltage and current of the PV module are determined as input to the control module. Then, the duty cycle is obtained through the control module. After that, it transmits the signal to IGBT of the DC–DC circuit. The following connection part is load. The control flow diagram is shown in Figure 5.

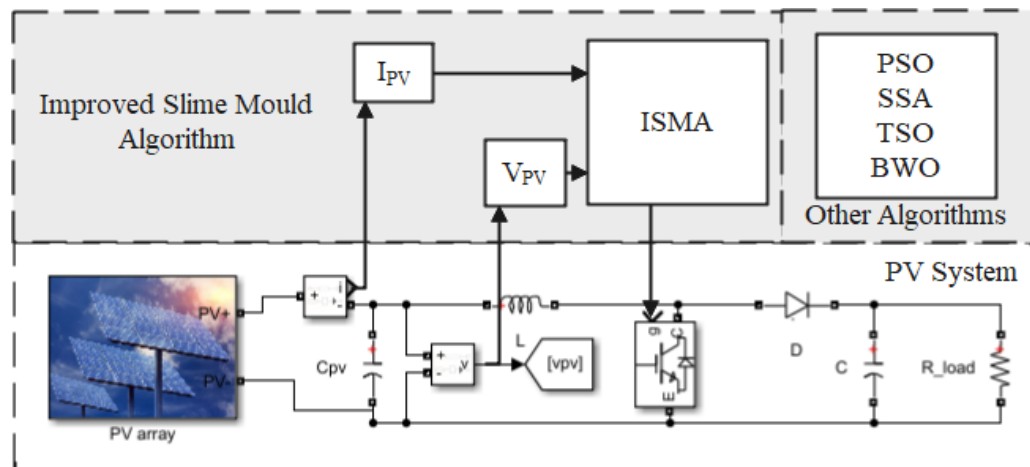

**Figure 5.** Control block diagram of the improved slime mold algorithm.

The output voltage value of the boost module can be calculated as: $V_o = I_o R$, where $V_o$ and $I_o$ are the output voltage and current, respectively; and the output current of the PV is $V_{pv}$ and $I_{ph}$. Disregard the power loss, the output power is equal to the input power, therefore, the output voltage can be expressed as follows:

$$V_o = \frac{1}{1-D} V_{pv} \tag{2}$$

where $D$ is the duty cycle, which can be obtained as follows:

$$D = 1 - \frac{V_{pv}(\min) \times \eta}{V_o} \tag{3}$$

where $\eta$ is the efficiency of the boost converter, a certain power loss must exist in the PV generation process; and $V_{pv}$ (min) is the minimum input voltage of the PV system. Relevant parameters of the mathematical model of photovoltaic power generation and DC–DC circuit are illustrated in Table 1.

**Table 1.** Parameters of mathematical model of photovoltaic and DC–DC circuit.

| PV Module | | DC–DC | |
|---|---|---|---|
| $I_L$ (A) | 7.8649 | $C_1$ (F) | $500 \times 10^{-6}$ |
| $I_o$ (A) | $2.9259 \times 10^{-10}$ | $L$ (H) | 0.0086 |
| $R_{sh}$ (Ω) | 313.3991 | $C_2$ (F) | $2.03 \times 10^{-5}$ |
| $R_s$ (Ω) | 0.3938 | $R_L$ (Ω) | 20 |

## 3. Model Construction of GMPP Control Algorithm Based on the Improved Slime Mold Algorithm

### 3.1. Inspiration

The slime mold algorithm (SMA) is an optimization algorithm based on slime mold feeding behavior. In the feeding process, slime mold possesses the property of contracting. Additionally, the slime mold may search for unknown areas when acquiring food sources. The behaviors of slime mold can be summarized as approaching food, wrapping food, and acquiring food.

### 3.2. Approaching Food

The convergence behavior of slime mold can be described as a mathematical model, and the contraction pattern of slime mold can be illustrated as:

$$X(t+1) = \begin{cases} X_b(t) \times vb \times (W \times X_A(t) - X_B(t)), r < p \\ vc \times X(t), r \geq p \end{cases} \quad (4)$$

where $X(t)$ denotes the current position of the slime mold, $vc$ is a random number between $[-1, 1]$, $X_A(t)$ and $X_B(t)$ are two random individuals, $X_b(t)$ is the individual with optimal adaptation for the current number of iterations, $t$ represents the current number of iterations, $w$ is the weight coefficient, and $vb$ denotes the random individual between $[-a, a]$.

$$p = \tanh|S(i) - DF| \quad (5)$$

where $i$ belongs to $[1, n]$, $S(i)$ denotes the adaptation value of the lower $i$ Mucor individuals, and $DF$ is the individual with optimal adaptation of the population.

$$a = \text{arctan}h(-(\frac{t}{T}) + 1) \quad (6)$$

$$W(SI(i)) = \begin{cases} 1 + r \times \log(\frac{bF - S(i)}{bF - wF} + 1), Pr \\ 1 - r \times \log(\frac{bF - S(i)}{bF - wF} + 1), Ot \end{cases} \quad (7)$$

$$SI = sort(S) \quad (8)$$

where $Pr$ is the individuals ranked in the first half, $Ot$ is the remaining individuals, $r$ represents a random number between $[0, 1]$, $wF$ denotes the worst fitness value, $bF$ is the best fitness value for the current number of iterations, and $SI(i)$ is the sequence of fitness, which indicates the increasing series.

### 3.3. Wrapping Food

The equation for updating the position of individual slime mold can be illustrated as follows:

$$X(t+1) = \begin{cases} rand \times (UB - LB) + LB, rand < z \\ X_b(t) + vb \times (W \times X_A(t) - X_B(t)), r < p \\ vc \times X(t), r \geq p \end{cases} \quad (9)$$

where $z$ is the bounded value (0.03), $rand$ means a random number between $[0, 1]$, and $UB$ and $LB$ are the upper and lower bounds of the current number of iterations, respectively.

### 3.4. Acquiring Food

The value of $vc$ oscillates between $[-1, 1]$ and the value of $vb$ is chosen randomly between $[-a, a]$, eventually converging to 0. The formula can be given as follows:

$$vc = [-b.b] \quad (10)$$

$$b = 1 - \frac{t}{T} \quad (11)$$

The flow of the algorithm can be demonstrated as follows:

- Initialize the population and set the corresponding parameters;
- Calculate the fitness values and sort them;
- Update the population position;
- Calculate the fitness value and update the optimal position;
- If the output performance indicators do not meet the requirements, steps 2–5 would be repeated.

### 3.5. Improved Slime Mold Algorithm in Shaded Photovoltaic System

The improved slime mold algorithm can be described as a nonlinear decreasing strategy, proposed by modifying the formula of parameter *a*. With the trough decreasing slowly in the early stage and speeding up in the later stage, the convergence speed of the system can be improved. However, the maximum power tracking error may still exist.

The algorithm is further optimized. All individuals are first rearranged according to their fitness, and then, the individuals with better fitness are selected through the linearly decreasing selection range. Additionally, a spiral search strategy is added based on the original slime mold search, which expands the algorithm's search range and improves the global search performance. The control flow chart of the improved algorithm is demonstrated in Figure 6.

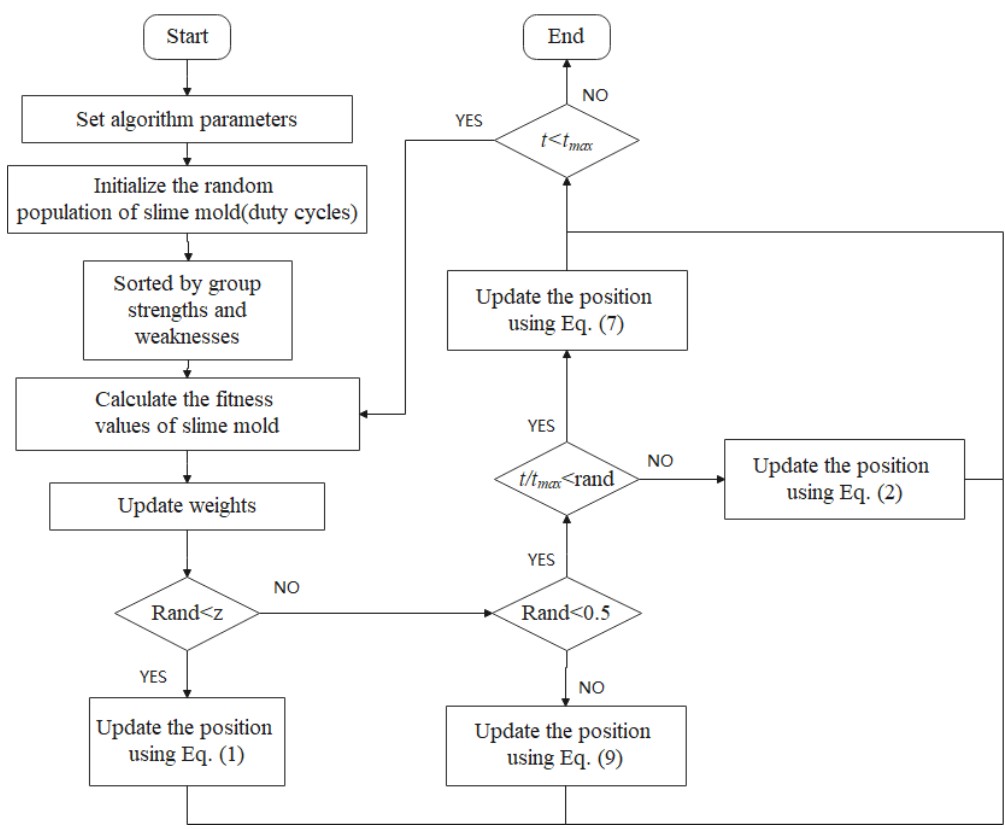

**Figure 6.** Flow chart of the improved slime mold algorithm for optimal MPPT control.

### 4. Simulation Analysis

Five algorithms—PSO, SSA, TSO, BWO, and ISMA—were built in this section. Many of the parameters of several of the algorithms were kept consistent to compare their performance. The parameters of the PV module were determined consistently for all algorithms, applying three PV modules in series, labeled as 1, 2, 3, with each comprising 20 individual PV modules. The relevant parameters of the PV module are listed in Table 2.

**Table 2.** Photovoltaic module related parameters.

| PV Module | Lsoltech | LSTH-215-P | |
|---|---|---|---|
| $P_{max}$ (W) | 213.5 | $I_m$ (A) | 7.35 |
| $V_{oc}$ (V) | 36.3 | Temperature coefficient of $V_{oc}$ (V/°C) | −0.36099 |
| $V_{max}$ (V) | 29 | Temperature coefficient of $I_{sc}$ (A/°C) | 0.102 |
| $I_{sc}$ | 8.37 | Cells of per module | 60 |

In summary, assuming the outside temperature of the PV panel is 25 °C, the PV shading condition is divided into the following three modes:

- Mode 1: PV panel radiance is 1000/400/800 W/m$^2$, the global maximum output power of PV power generation in this mode is 5759 W;
- Mode 2: PV panel radiance is 300/500/700 W/m$^2$, through observing the P–V characteristic curve, we see the global maximum power point is 2860 W;
- Mode 3: The radiance is set to 200/400/600 W/m$^2$, this case simulates the power generation process with a large shading area, the global maximum power tracking point is 2253 W.

The P–V and I–V curves for the three shaded PV cases are shown in Figure 7.

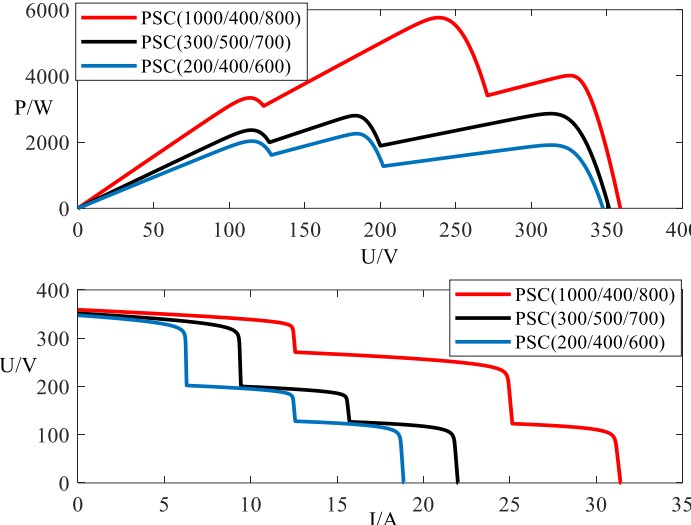

**Figure 7.** PV output characteristic curves under three different shading conditions.

In the first shading condition, as shown in the Figure 8, all algorithms tracked the maximum power point. The response time of ISMA was the shortest. After 0.2010 s, ISMA reached a steady-state. PSO had a tracking time of 0.2312 s. The response time of the other three algorithms (SSA, TSO, and BWO) were 0.3514 s, 0.3129 s, and 0.2424 s, respectively. From the power performance index curves of the five algorithms, all tracked the maximum power (5761 W). Compared with SSA, ISMA's tracking time was reduced by 42.8%. ISMA only had a small amount of power fluctuation compared with SSA/TSO/BWO. Significant power fluctuations during the initial 0.6 s existed in the control process of PSO. From the duty cycle modulation curves, it was observed that PSO and SSA displayed significant power oscillations, which were caused by the updating method. While other controls possessed a fast curve adjustment process, the ISMA algorithm, in particular, remained fixed in all iterations.

In the second shading condition (PSC2), the value of maximum power was 2860 W. The simulation results of the five algorithms are displayed in Figure 9. As shown in the figure, PSO and BWO could not track the power point of the PV system. Both fell into the local maximum power point. There were minor differences between SSA, TSO and ISMA. A 99.3% tracking efficiency was achieved by ISMA, which possessed the shortest response time. The final power obtained by the PSO, SSA, TSO, BWO, and ISMA algorithms, was 2438 W, 2832 W, 2650 W, 2825 W, and 2840 W, respectively, and the response time was 0.2256 s, 0.4222 s, 0.3403 s, 0.2092 s, and 0.1899 s, respectively. The tracking effect of BWO and TSO was poor. The other three algorithms tracked significantly. The ISMA algorithm, in particular, achieved 99.3% tracking efficiency. In terms of response time, it showed a 55% reduction compared with the SSA algorithm. The ISMA algorithm eliminated individuals continuously with substandard adaptation values, and selected individuals randomly for varying updates according to the probability. It is worth mentioning that individual output parameters cannot reflect index comparison. Through analysis and comparison of the five different control methods, the result showed that the algorithm proposed in this article possessed fast response time and relatively fewer fluctuations.

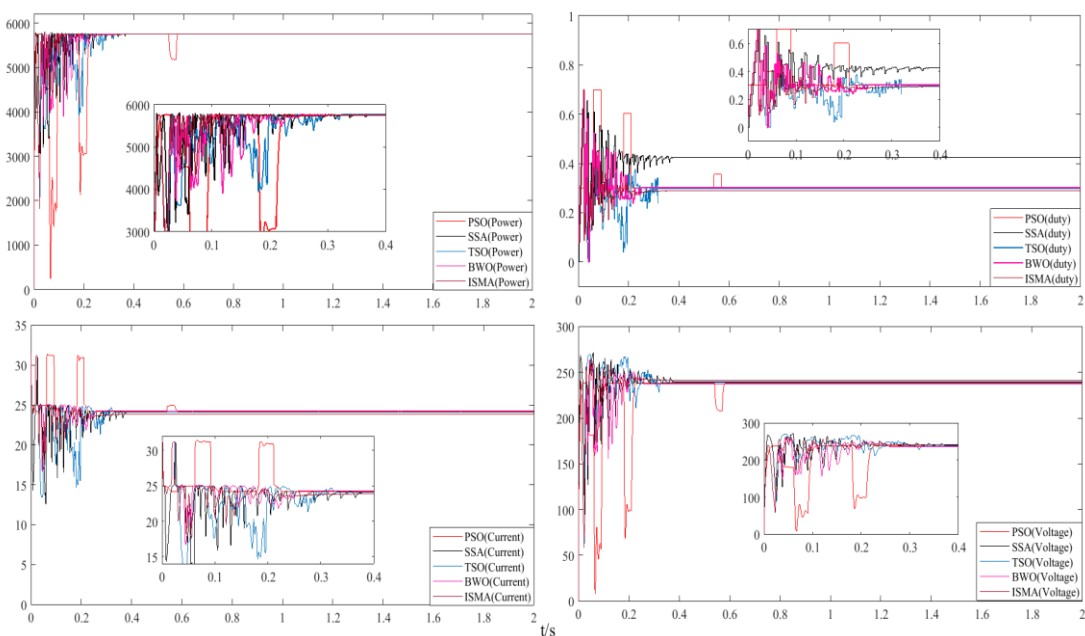

**Figure 8.** Output curves of five different algorithms under PSC1 conditions.

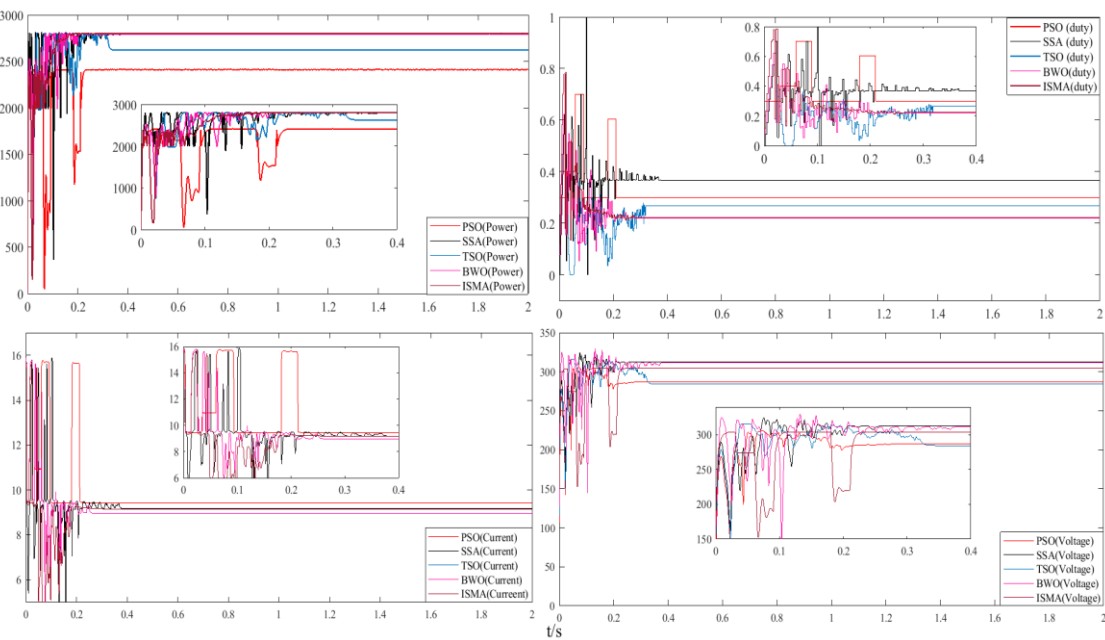

**Figure 9.** Output curves of five different algorithms under PSC2 conditions.

Under the third shading condition (PSC3), the maximum power was 2253 W. As shown in the Figure 10, both PSO and BWO algorithms are unable to track the maximum power point. They fell into two local maximum power points of 2029 W and 1872 W, respectively. There were minor differences between SSA, TSO and ISMA. A 100% tracking efficiency was achieved by ISMA. The three maximum power tracking values were 2252 W, 2251 W, and 2253 W, respectively. The response time of ISMA was 0.1901 s, followed by BWO with a response time of 0.2262 s. The response times of PSO, SSA, and TSO, were 1.09 s, 0.3821 s, and 0.296 s, respectively. Simulation outcomes demonstrated that the response time of ISMA was 82.5%, 50.2%, 35.8%, and 16% shorter, compared with other algorithms. The duty cycle curves between the different controls were compared and analyzed, showing that the PSO algorithm possessed poor modulation with many fluctuations, and the ISMA algorithm outperformed the other algorithms to some extent.

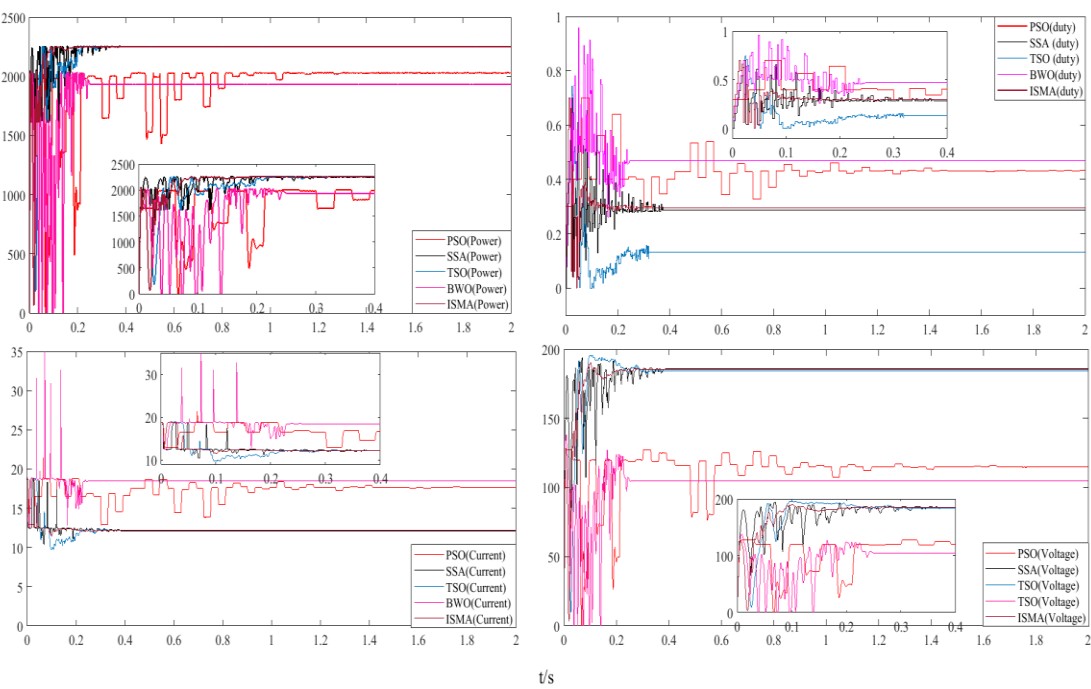

**Figure 10.** Output curves of five different algorithms under PSC3 conditions.

In summary, under a small area of shadow, the maximum power point could be tracked by all five algorithms. PSO existed with large fluctuations, and the ISMA algorithm possessed the best performance with a response time of 0.1901 s. The PSO and TSO algorithms failed when the difference between the local maximum power point and the global maximum power point was not significant, ISMA possessed the best output in all aspects, tracking efficiency to 99% or more. The SSA algorithm had the longest response time. Eventually, under large area shadow conditions, the PSO and BWO algorithms failed, and fell into the local maximum power point of 2029 W and 187 W, respectively. The other three algorithms were able to track the maximum power point, and the ISMA algorithm achieved 100% tracking efficiency. In terms of response time, ISMA decreased by 42.8%, compared with the SSA algorithm. The simulation consequences illustrate that the ISMA can track the maximum power of PV power generation under different shading conditions. The performance parameters of different algorithms are demonstrated in Table 3.

**Table 3.** Comparison of performance parameters of different control algorithms.

| Irradiance (W/m²) | Algorithm | $P_{max}$ (W) | $P_{pv}$ (W) | Tracking Speed (s) | MPPT Efficiency (100%) |
|---|---|---|---|---|---|
| PSC₁ (1000 400 800) | PSO | | 5760 | 0.23 | 99.98 |
| | SSA | | 5761 | 0.375 | 100 |
| | TSO | 5761 | 5761 | 0.3 | 100 |
| | BWO | | 5761 | 0.24 | 100 |
| | ISMA | | 5759 | 0.2010 | 99.97 |
| PSC₂ (300 500 700) | PSO | | 2438 | 0.255 | 87.07 |
| | SSA | | 2832 | 0.392 | 99.02 |
| | TSO | 2860 | 2650 | 0.304 | 92.66 |
| | BWO | | 2825 | 0.2105 | 98.8 |
| | ISMA | | 2840 | 0.1899 | 99.3 |
| PSC₃ (200 400 600) | PSO | | 2029 | 1.07 | 90.06 |
| | SSA | | 1872 | 0.27 | 83.09 |
| | TSO | 2253 | 2250 | 0.36 | 99.87 |
| | BWO | | 2250 | 0.22 | 99.87 |
| | ISMA | | 2253 | 0.1901 | 100 |

## 5. Experimental Verification

All algorithms were validated through the HIL + RCP platform. Yuankuan Energy Technologies Ltd. has launched a real-time simulation platform, known as being configurable. By downloading the model, HIL can map with the I/O of the hardware. To achieve self-closing of the application model's sub-module for this research, a custom control block was run on the CPU. Importing PV CELL module into the Control Block, likewise, importing DC–DC circuit into the FPGA. Finally, the duty cycle module delivered a pulse to the IGBT. The schematic diagram of the operation of the HIL is illustrated in Figure 11.

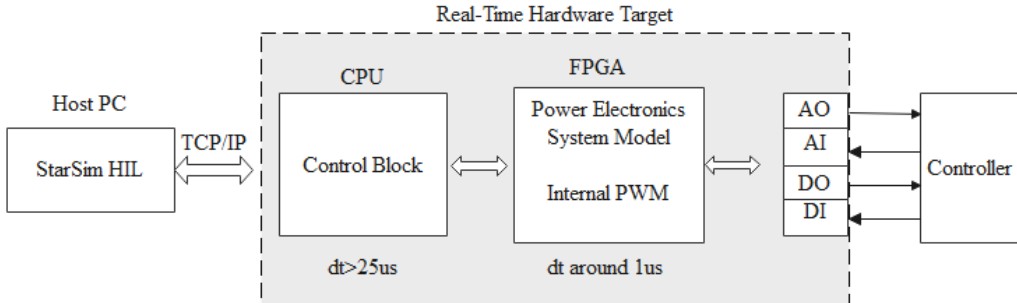

**Figure 11.** Schematic diagram of HIL.

StarSim RCP included a host part and a real-time part, which imported the control part into the RCP through the running of a computer program. The algorithm was imported into the RCP module, generating pulses via PWM, to achieve closure between HIL and RCP. The topology of the HIL + RCP module is described in Figure 12.

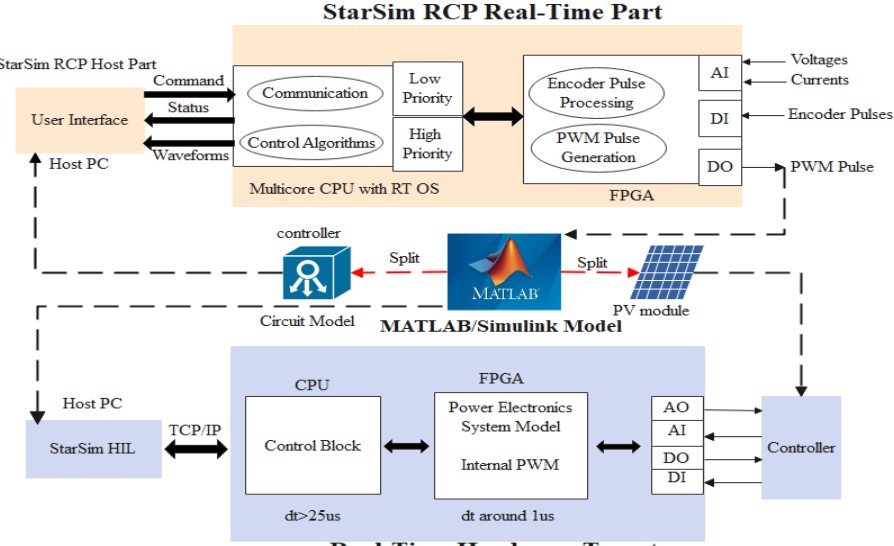

**Figure 12.** Schematic diagram of HIL + RCP.

The PV in MATLAB/Simulink could not be recognized in the experiment. A specific PV module was selected. Considering the fluctuation of PV output parameters was large, a small amount of PV panel would be severely affected by harmonics. Therefore, a large number of PV cells were applied, by connecting the PV cells in series and parallel (35 in series and 20 in parallel). Two PV cell modules of this scale were selected, one with radiance of 1000 W/m$^2$ and the other with radiance of 500 W/m$^2$. The output P–U and I–U characteristic curves of the photovoltaic system are shown in Figure 13. The relevant parameters of applied PV cells are listed in Table 4. The platform equipment graphic is displayed in Figure 14.

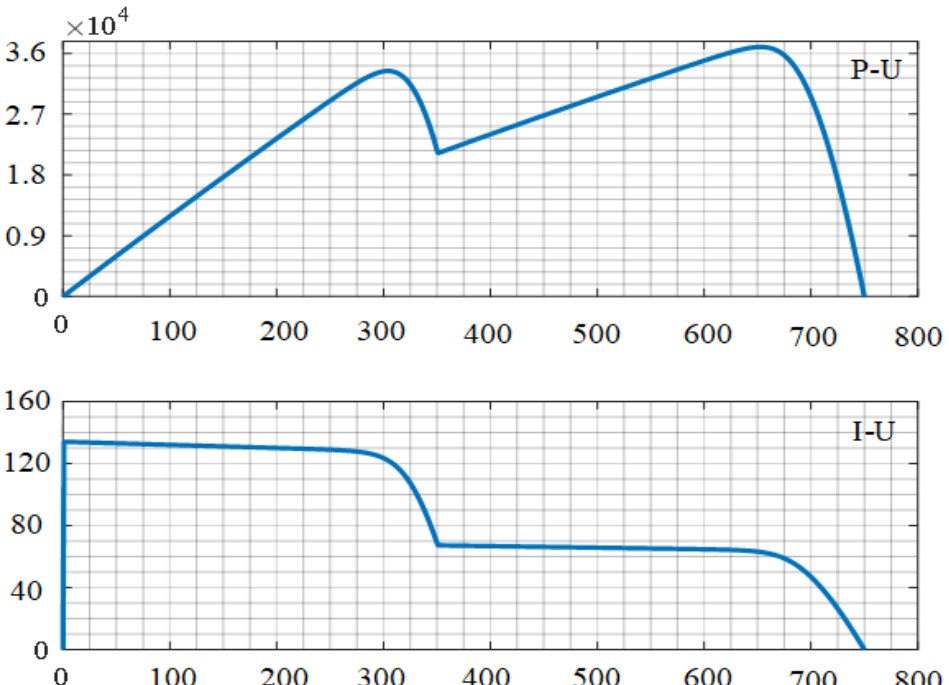

**Figure 13.** P–U and I–U characteristic curves.

**Table 4.** Parameters related to the photovoltaic cell for the experiment.

| PV Module | | | |
|---|---|---|---|
| $I_{sc}$ (A) | 3.35 | Radiance (W/m$^2$) | 500/1000 |
| $I_m$ (A) | 3.05 | Series cells | 35 |
| $V_{oc}$ (V) | 21.7 | Parallel cells | 20 |
| $V_m$ (V) | 17.4 | T (°C) | 25 |

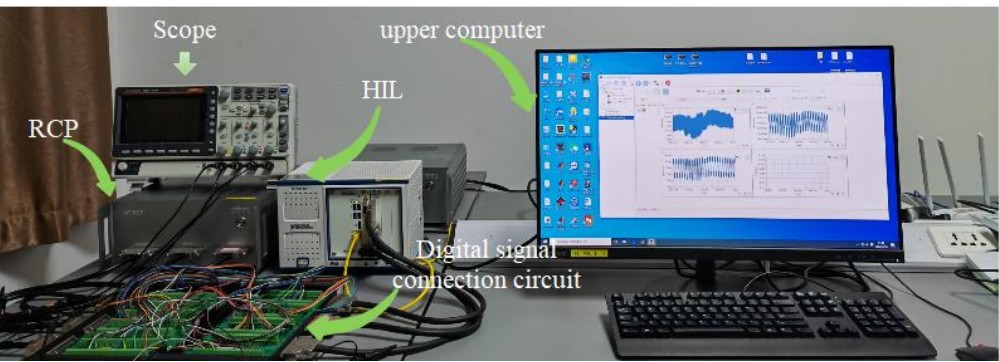

**Figure 14.** Platform diagram.

Considering the waveform output threshold range of the RCP, the voltage was reduced by a factor of 100, and the current by a factor of 10. The experimental graphs of the five algorithms are shown as follows (Figures 15–19). All data were from real-time tests. In addition, the time range for the platform's output data, is randomness. Therefore, after running for 10 s, the enable module switch was turned on.

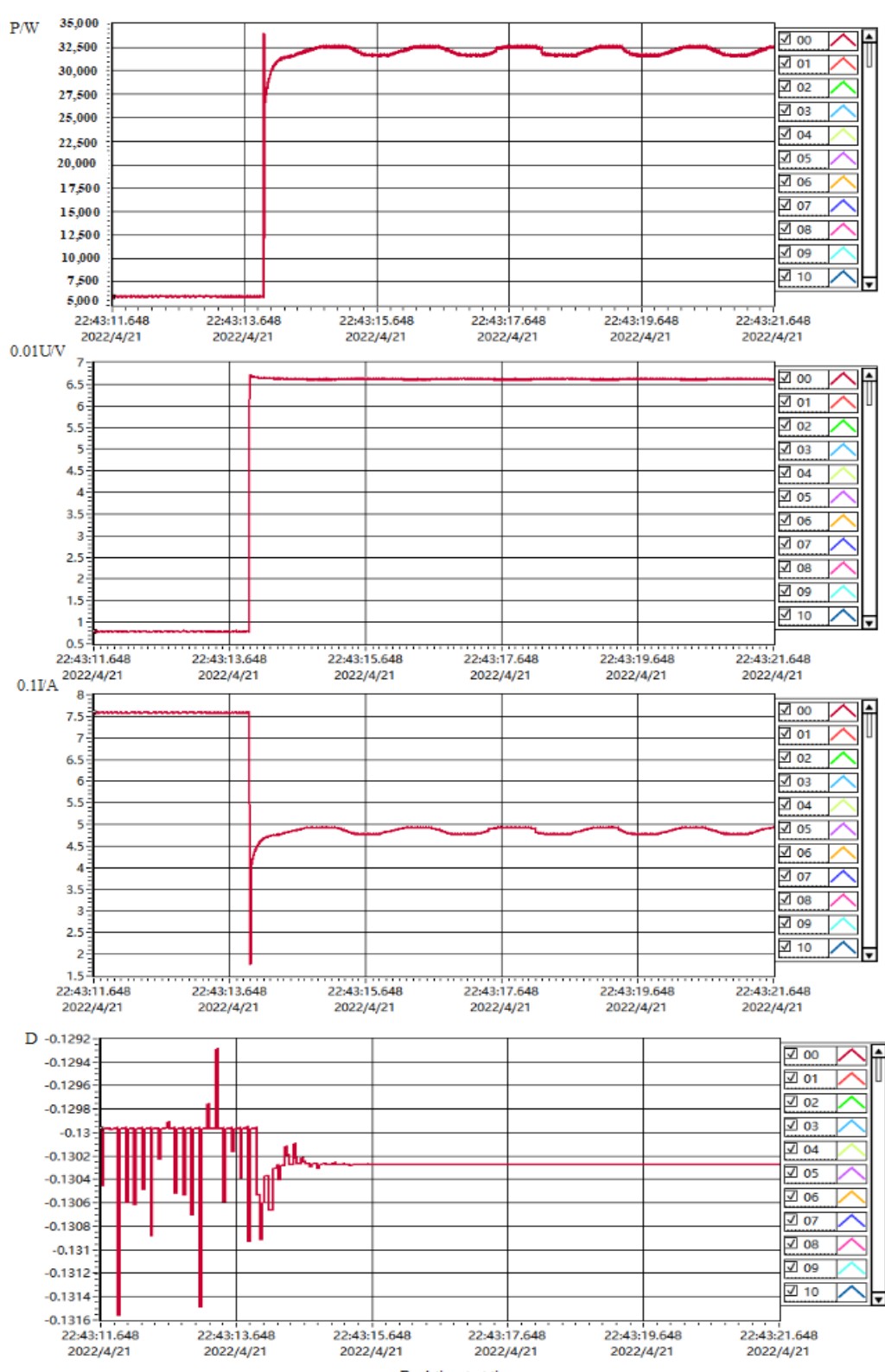

**Figure 15.** PSO output curves based on real-time simulation platform.

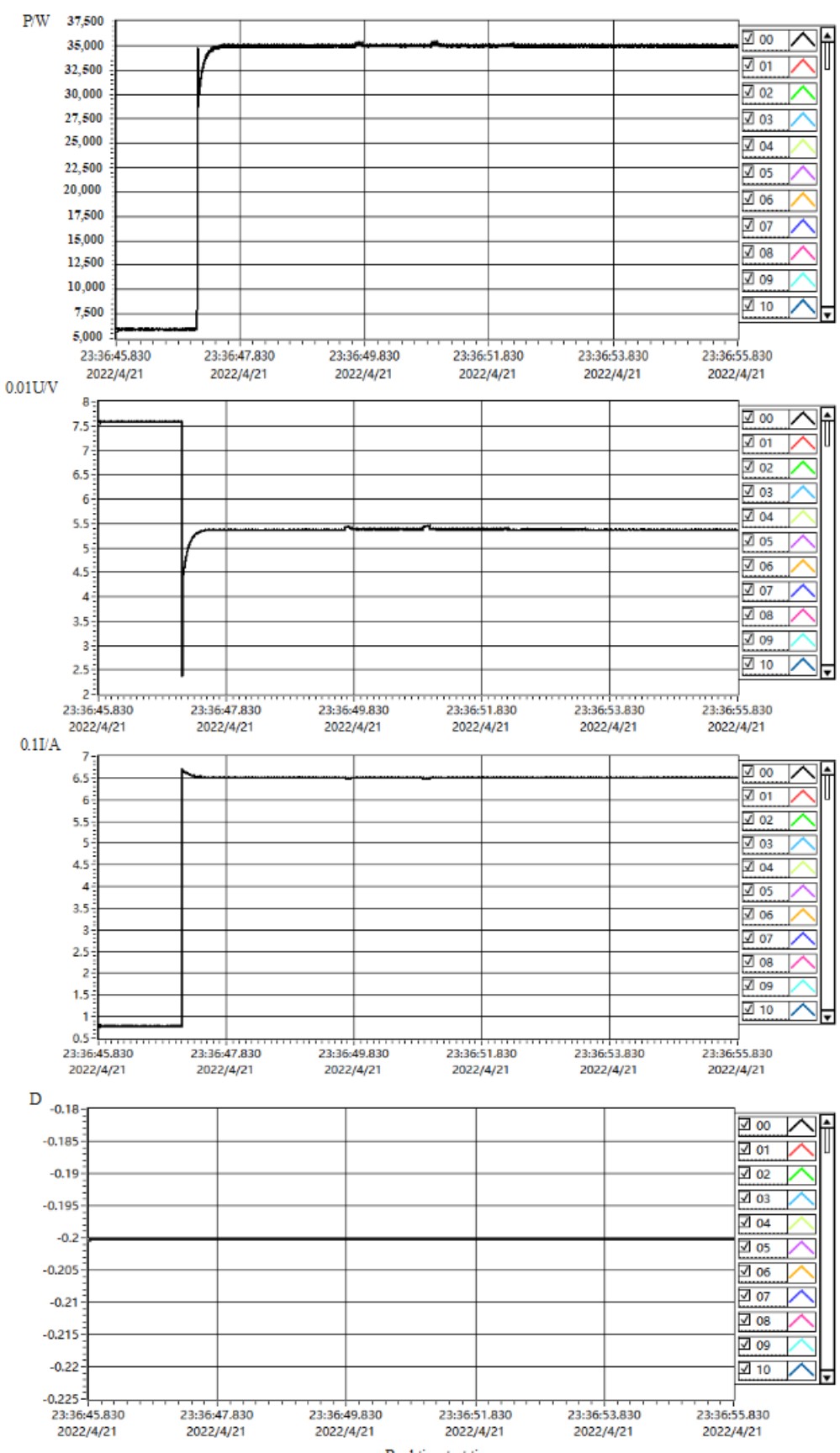

**Figure 16.** SSA output curves based on real-time simulation platform.

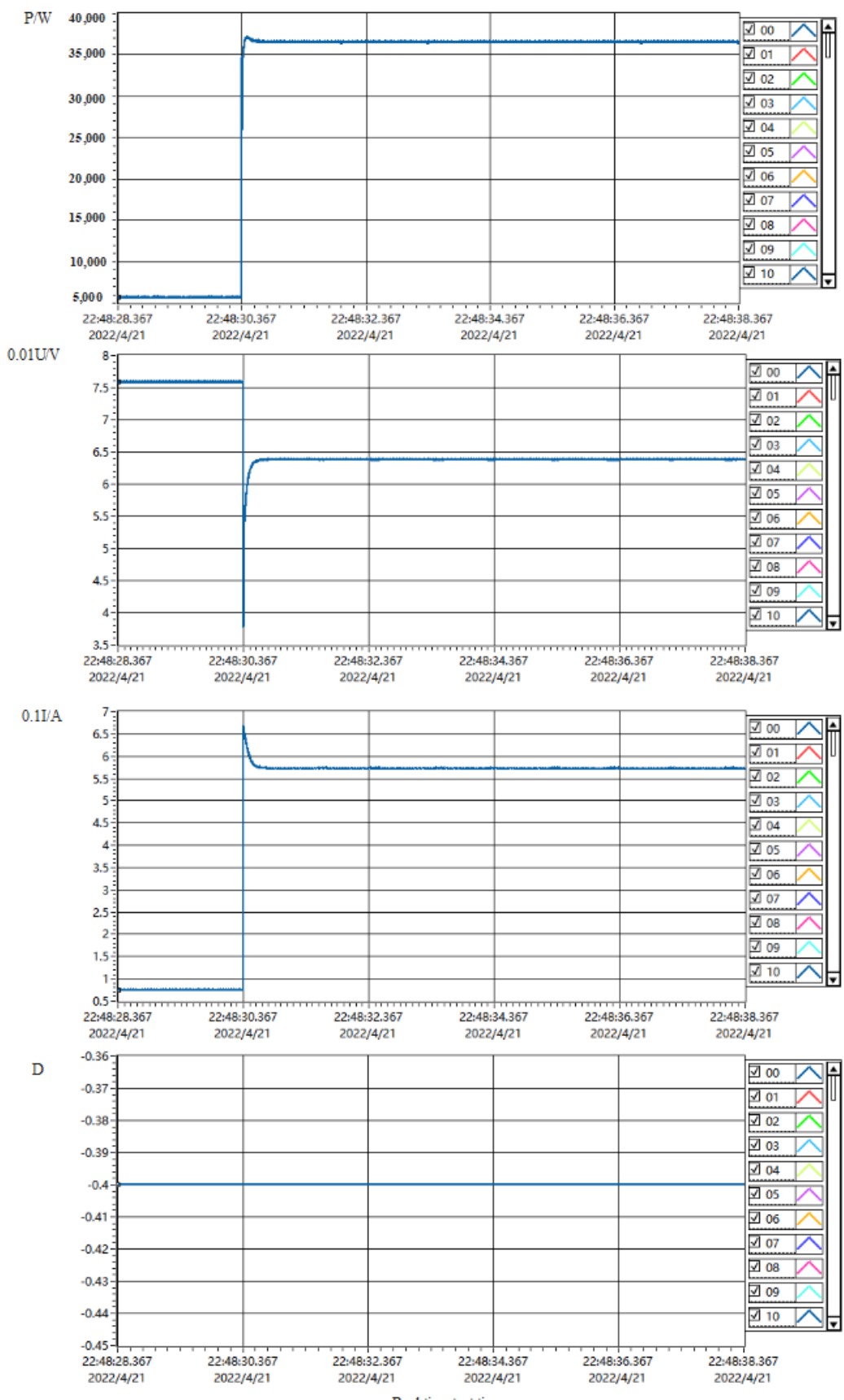

**Figure 17.** TSO output curves based on real-time simulation platform.

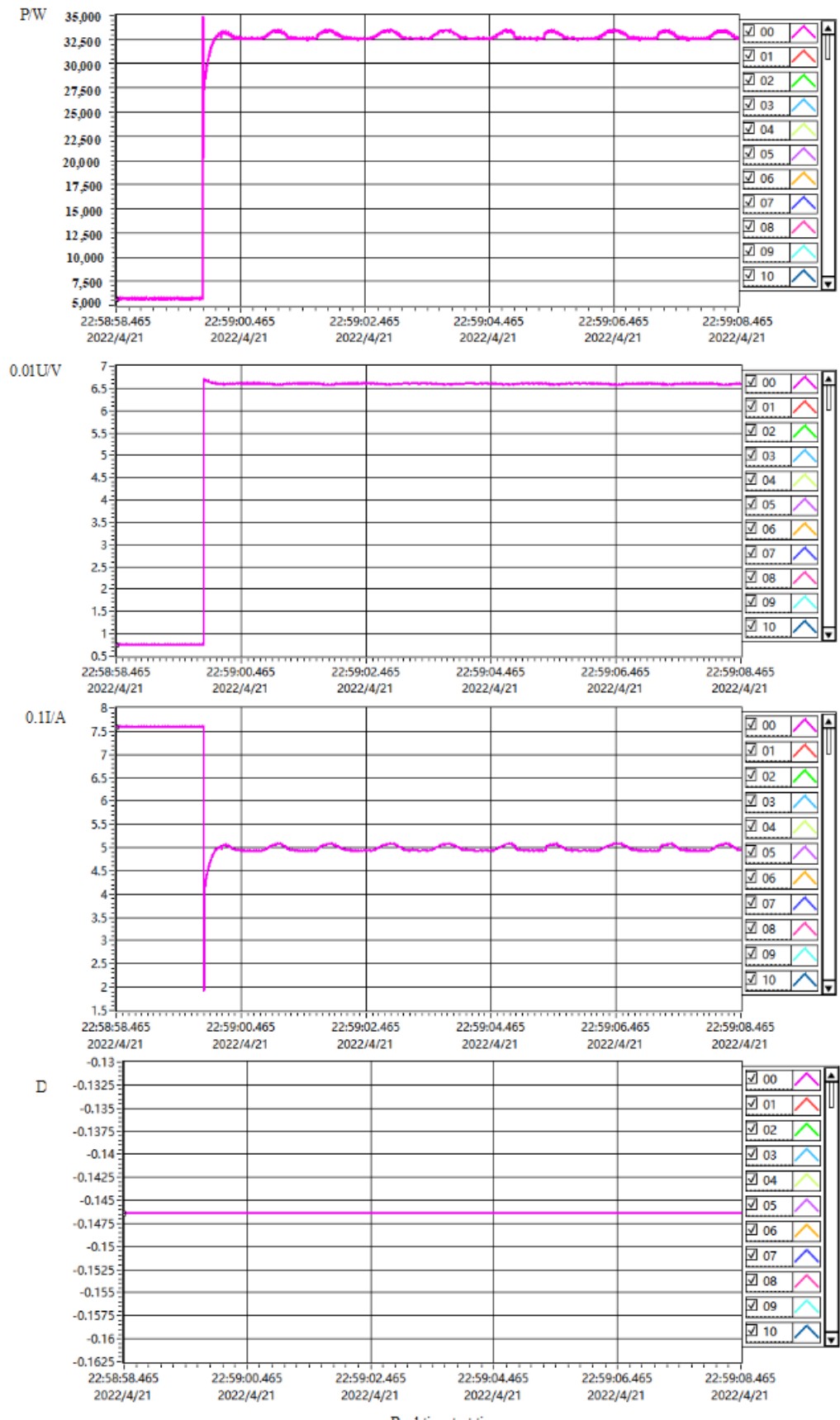

**Figure 18.** 1BWO output curves based on real-time simulation platform.

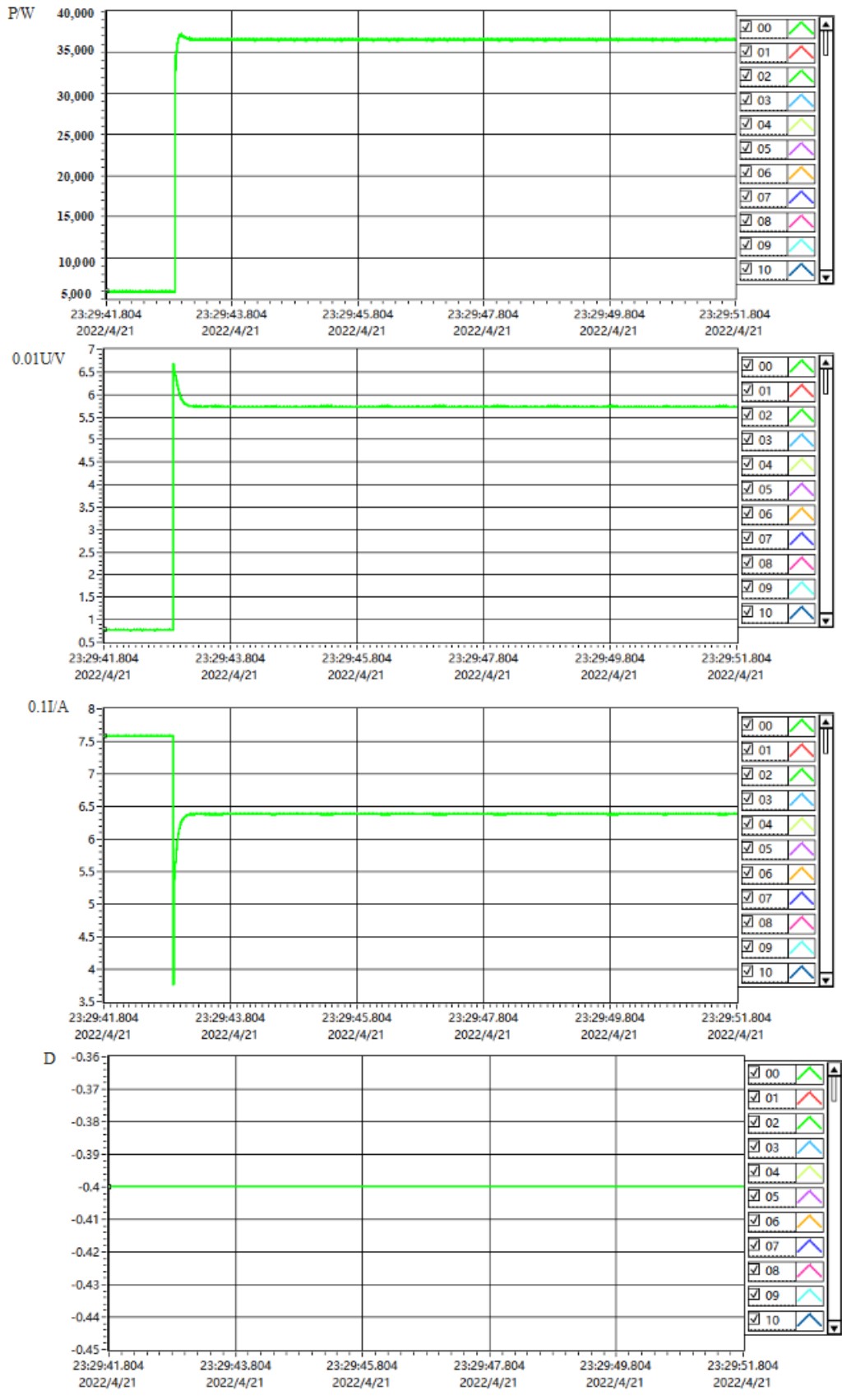

**Figure 19.** ISMA output curves based on real-time simulation platform.

It can be seen from Figures 15–19, that the PSO algorithm track failed. The algorithm eventually fell into a local maximum power point of about 32,521 W. In addition, the PSO algorithm had large fluctuations. The other four algorithms, under this shading condition, were able to track the global maximum power point, however, there was a large gap in tracking efficiency. The tracking power of the SSA algorithm was 35,002 W, and a power fluctuation of about 300 W existed in the middle process (verified by several tests). The maximum power of the TSO algorithm was 36,431 W, with good uniformity of the regulation process, small power fluctuation and slight jitter. The efficiency of the SSA algorithm reached more than 94%. In addition, the TSO algorithm tracked well under the shading conditions of the experiment, whose efficiency reached more than 94%. The tracking power of the BWO algorithm was 33,466 W, which was similar to the PSO algorithm, however, the regulation process was prone to regulation failure due to the defects of the population-seeking approach. The ISMA algorithm, proposed in this research, tracked the maximum power (36,692 W) and less than 1% error occurred. The power curve was smooth and free from power fluctuations. The performance parameter about the five methods are shown in Table 5.

**Table 5.** Comparison of performance parameters of different control algorithms.

| Irradiance (W/m$^2$) | Algorithm | $P_{max}$ (W) | $P_{pv}$ (W) | MPPT Efficiency (100%) |
|---|---|---|---|---|
| | PSO | | 32,521 | 87.89 |
| | SSA | | 35,002 | 94.6 |
| (500 1000) | TSO | 37,000 | 36,431 | 98.46 |
| | BWO | | 33,466 | 90.45 |
| | ISMA | | 36,692 | 99.17 |

## 6. Conclusions

An MPPT strategy based on the ISMA algorithm is proposed in this research. Five algorithms—PSO, SSA, TSO, BWO, ISMA—were constructed in a MATLAB/Simulink platform. The simulation consequences demonstrated that the ISMA algorithm possessed small fluctuation and rapid response. Under three different shading conditions, the tracking efficiency of the ISMA algorithm was 99.97%, 99.3%, and 100%, respectively. The response times of ISMA were short, all within 0.2 s. The HIL + RCP physical platform was applied to verify the adaptability of all algorithms. ISMA achieved a power tracking efficiency of 99.17%, being an increase of 11.28% compared with the PSO algorithm. The experimental outcomes illustrate that the ISMA algorithm has low fluctuation and a good tracking effect. In the future, our team will conduct research on the situation of time-varying PV shading and the effect of photothermal characteristics of PV cells under shading conditions.

**Author Contributions:** Conceptualization, C.F.; methodology, C.F.; software, C.F. and L.Z.; validation, C.F. and W.D.; formal analysis, C.F.; investigation, C.F. and L.Z.; resources, W.D.; data curation, C.F.; writing—original draft preparation, C.F.; writing—review and editing, C.F. and L.Z.; visualization, C.F.; supervision, C.F.; project administration, L.Z.; funding acquisition, L.Z. All authors have read and agreed to the published version of the manuscript.

**Funding:** This research supported by the Natural Science Foundation of China under grant no. 52065055, and Provincial and Ministerial Projects No. 2021JS004.

**Institutional Review Board Statement:** Not applicable.

**Informed Consent Statement:** Not applicable.

**Data Availability Statement:** Not applicable.

**Conflicts of Interest:** The authors declare no conflict of interest.

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
