# Peer review of "Research and Application of MPPT Control Strategy Based on Improved Slime Mold Algorithm in Shaded Conditions"

_electronics, doi:10.3390/electronics11142122_

Round 1

Reviewer 1 Report

Strengths:

·         The MPPT tracking algorithm based on the bionic algorithm has higher tracking efficiency, shorter response time, and better flexibility and adaptability compared to traditional techniques.

·         Superior performance of slime mould algorithm under PV solar cell shaded conditions vis a vis similar population-based meta-heuristic algorithms in terms of short response and better accuracy.

·         Further modifications are made to strike a balance between the exploration and exploitation phase and account for accelerated global optimum search, convergence speed, and solution accuracy.

·         Selection of swarm-based algorithms over evolutionary algorithms for the reasons such as less number of operators to be controlled and better utilization of historical information.

·         ISMA seems to optimally avoid convergence at local maxima power peaks.

·         ISMA is mentioned to be tracking MPP in all shading conditions [PSC1, PSC2 & PSC3] with fewer iterations and quicker response.

·         Development of closed-loop feedback intelligent control for better stability.

·         Without the adaptive weight modification on the fly, all the meta-heuristic algorithms selected in the research study are highly likely to suffer from local minima peak traps and low convergence rate in the iterative process except ISMA.

·         Good representation of data and plots with results to confirm simulation vis a vis experiment in all the modes of testing conditions.

·         Good configuration of HIL and RCP setup to test the theory.

·         Optimal selection of PV shaded modes to utilize multi-model functions.

Challenges: Accuracy, response time, degree of randomness, and order of complexity

·         Degree of randomness in the exploration phase of solution search space accounted for in all five meta heuristic swarm algorithms.

·         Order of complexity vis a vis number of input parameters in a PV system.

·         Mathematical modeling of the interface between electrical and thermal characteristics of PV system, which are often non-linear and interdependent.

Weakness:

·         The shadow cast on a panel is uniform throughout the surface and eliminates the possibility of variation in thermal propagation.

·         The relevant experiment study with RCP + HIL seems to be conducted at only one mode of PV shaded condition.

·         The PSO algorithm is highly susceptible to particle topology, and therefore the merits of ISMA selection over PSO are not absolute.

·         Under mild shadow conditions, the domination of ISMA seems not really significant.

·         The transient profile of the duty cycle estimated by ISMA seems to contribute more to switching losses (IGBT triggering) and thereby might increase an overall loss component in the PV system.  

·         Power loss components in both PV panel and DC-DC converter are not taken into account either in simulation or RCP/HIL studies.

Improvements:

·         Section 2.2 requires reexamining on the definition of the parameters in the PV model equivalent circuit model as there is an inconsistency between the circuit model and parameters mentioned in the formula [ For example, “I” is called as shunt current instead of PV panel output current].

·         Also, similar care should be taken in section 2.3 while deriving expression for   PV panel output current.

·         The poor-performing pair of algorithms in terms of tracking MPP in section 4 [PSC 2] are PSO & TSO but not PSO & BWO.

Conclusion:

1.      The paper verifies the superiority of the ISMA in tracking MPP with shorter response and fewer iterations compared to similar swarm algorithms in a controlled multi solar irradiance environment.

2.      Both the simulation and HIL studies confirm the better performance of ISMA over rest of algorithms.

3.      Good use of data and plots to justify the performance of ISMA in a PV system MPPT study.

4.      Including a few more PV shade conditions in HIL + RCP test would attest the simulation results better with experiment.

5.      Consideration of loss components from PV system in control strategy would further consolidate the research study scope.

6.      The initial transient profile of duty cycle would be a principal component of switching loss in a real time dynamic solar irradiance environment.

7.      ISMA seems a clear choice of algorithm among population-based search algorithms especially in extreme PV panel shaded environment.

8.      Overall, the paper is successful in highlighting the performance of ISMA and justifies its application superiority in PV shaded condition through simulation and experiment.

Author Response

First, we would like to thank the reviewers for their recognition of our work.

  1. In response to the reviewers' challenge section. Our response is as follows.

(1). The optimization search of the bionic algorithm does have a large randomness, and the simulation of this study is tested through several simulations to continuously adjust the parameters of the algorithm and get more stable control results

(2). For the radiance as well as the complexity of the PV system. This is the direction of our team's subsequent preparation for research. For the photothermal characteristics of the PV system during actual operation, this is also one of our future research directions, and the authors have included this part in their future plans for the paper. Once again, we thank the reviewers for their proposal.

In response to the weaknesses section, our response is as follows.

 Q1:The shadow cast on a panel is uniform throughout the surface and eliminates the possibility of variation in thermal propagation.

 Q2:The relevant experiment study with RCP + HIL seems to be conducted at only one mode of PV shaded condition.

We will perform algorithm validation on actual PV power systems in the future, and the actual algorithm will be artificially set to multiple groups for the shaded condition. We hope the reviewers understand that we will focus on different shading conditions and time-varying shading in the future. Thanks again to the reviewers.

Q3:The PSO algorithm is highly susceptible to particle topology, and therefore the merits of ISMA selection over PSO are not absolute.

Q4:  Under mild shadow conditions, the domination of ISMA seems not really significant.

Under mild shadow conditions, the control algorithm proposed in this study is indeed not obvious enough. Considering the actual photovoltaic power generation, the shadow conditions cannot be predicted, therefore, the control algorithm proposed in this study is highly adaptable by combining the shadow conditions of various cases. The traditional PSO algorithm position update method is superior to certain limitations, while the slime foraging algorithm of this study has a better search mechanism. If the PSO algorithm is improved, it may back change the output performance of PSO algorithm, and we have been committed to research and improve various intelligent algorithms. We hope to invite reviewers again to review our research results.

 Q5: The transient profile of the duty cycle estimated by ISMA seems to contribute more to switching losses (IGBT triggering) and thereby might increase an overall loss component in the PV system.  

  • Q6:Power loss components in both PV panel and DC-DC converter are not taken into account either in simulation or RCP/HIL studies.

Both from the experimental and simulation results, the ISMA algorithm is able to track the maximum power quickly and maintain the largest power value among all algorithms. Therefore, it is enough to prove the superiority of the algorithm. According to the duty cycle curves of the five algorithms, ISMA has less fluctuation and the shortest response time, therefore, the algorithm has smaller power loss. The current HIL+RCP testbed compares the five control algorithms under the same conditions, and the test results show that the ISMA control is the best under the same conditions.

Improvements:

Q1:Section 2.2 requires reexamining on the definition of the parameters in the PV model equivalent circuit model as there is an inconsistency between the circuit model and parameters mentioned in the formula [ For example, “I” is called as shunt current instead of PV panel output current].

Q2: Also, similar care should be taken in section 2.3 while deriving expression for  PV panel output current.

Thank you to the reviewers for their questions. We have revised the figure in this section and the result of the revision is shown below. In addition, also in the later sections, we consider the consistency of the same parameters, etc. Thanks again. 2.1 and 2.2 have been combined into one section.

Q3:The poor-performing pair of algorithms in terms of tracking MPP in section 4 [PSC 2] are PSO & TSO but not PSO & BWO.

This is an oversight on our part. We have corrected that section, thank you.

Reviewer 2 Report

This paper proposed improved Slime Mould Algorithm to track the global peak power of the photovoltaic energy system under partial shading conditions.

The reviewer invites the authors to reply and take these comments into consideration as follow:

1.     In the second section, the authors focused more on the partial shading problem and PV model while small subsection (2.3) for system description. The reviewer thinks that there is no need to the first and second subsection (2.1 and 2.2) and the authors should focus and make the second section for: 2. Description and Modeling of the proposed PV System. This section should explain and describe the proposed PV system (power and control circuits). Power circuit includes the PV modules with clarifying its characteristics/parameters (in table), DC-DC converter, and load. The control circuit include proposed MPPT with demonstrating inputs and output.

2.     In Figure 5, the photo of the DC-DC converter is wrong. The used photo for power transformer AC-AC not the DC-DC converter.  The authors should modify and improve this figure totally.

3.     For PSC2(300/500/700), the global peak power seems to be allocated at the end of the P-U curve (Fig. 7) with a value 2800 W and VGPP=311 V not 190 V as obtained in Figure 9.  Also, the current at global peak power IGPP is 9 A not 15 as obtained in Figure 9.

4.     The declaration of figure 7 has been cited and appeared after the figure itself. It should be before the figure itself.

5.     For figures 8,9, and 10, It is better to merge the PV power for the five MPPT at same figure to be easier for the comparison purpose. Also, please do the same suggestion for PV current, voltage and duty cycle.

6.     In table 2, It is better to calculate the energy efficiency instead of the power efficiency.

7.     The authors applied their study under static partial shading condition, PSC1 or PSC2 or PSC3 separately. What is the response of the MPPT techniques if they applied dynamic partial shading conditions? This will discriminate and differentiate between all five MPPT algorithms clearly where some MPPT algorithm succeeded to track the global peak (GP) under PSC1 for example.

8.     It is better to apply different partial shading conditions with different position (PSC1: GP at the beginning, PSC2: GP at the middle, GP at the end) and values of global peak power.

9.     The simulation results have no additional contributions, and less value analysis and discussion (Line 377-381). The P-U curves of partial shading conditions under study in the experimental results should be introduced and the experimental results (Theoretical and actual) should be discussed and analyzed deeply.

10.  There are some existing studies applied the Slime Mould Algorithm to track the global peak under partial shading conditions like*. Therefore, the authors should review and add all existing studies related to the Slime Mould based MPPT Algorithm versions, provide the research gaps in the previous studies and highlight the improvement and contributions done for the proposed Slime Mould Algorithm.

*Mirza, A. F., Mansoor, M., Zhan, K., & Ling, Q. (2021). High-efficiency swarm intelligent maximum power point tracking control techniques for varying temperature and irradiance. Energy228, 120602.

11.  In abstract, the authors said, “The experimental outcomes show that the shaded PV global maximum power tracking control strategy based on the improved Slime Mould algorithm has superior performance indexes of the tracking process and effectively reduces the power loss in the PV power generation process”, this is not clear and have not been discussed clearly in the experimental results where there are some techniques has similar results to ISMA.

12.  There are some minor errors that should be modified as follow:

§  The caption of Table 1 and 2 was wrong. For example, the authors said: “Table 1. This is a table. Tables should be placed in the main text near to the first time they are cited”.

§  Line 248, the authors should discuss and compare the global power values for PSC2 where the authors compared the tracking time values only. What are the MPPT algorithms succeeded, and which failed to track the global peak.

§  Line 266, middle of the curve with a value of 2800 $W$. Please modify the unit to be W.

§  Line 267, revealed in Fig. Which Figure?

§   Line 286, the maximum power is 2253W not 2523w.

§   Line 289 with a response time of 0.2262 s not $s$.

§  Line 377, The experimental verification in Section V. please modify Section 5 not V

§  Avoid lumping many references together like [1]-[5] and [6]-[10] in the introduction section.

Author Response

Q1:   In the second section, the authors focused more on the partial shading problem and PV model while small subsection (2.3) for system description. The reviewer thinks that there is no need to the first and second subsection (2.1 and 2.2) and the authors should focus and make the second section for: 2. Description and Modeling of the proposed PV System. This section should explain and describe the proposed PV system (power and control circuits). Power circuit includes the PV modules with clarifying its characteristics/parameters (in table), DC-DC converter, and load. The control circuit include proposed MPPT with demonstrating inputs and output.

Thank you for your comments. We have merged the two parts of the previous section and added new tables(Table 1). The table contains the PV model and the relevant parameters of the DC-DC circuit.

Q2:In Figure 5, the photo of the DC-DC converter is wrong. The used photo for power transformer AC-AC not the DC-DC converter.  The authors should modify and improve this figure totally.

This was an oversight on our part. Thank you for your reminder, we have made a change to Figure 5, which is shown in the figure below.(The online submission version does not recognize images, so we hope reviewers understand that we will place your response directly before the manuscript.)

Q3: For PSC2(300/500/700), the global peak power seems to be allocated at the end of the P-U curve (Fig. 7) with a value 2800 W and VGPP=311 V not 190 V as obtained in Figure 9.  Also, the current at global peak power IGPP is 9 A not 15 as obtained in Figure 9.

Thanks to the reviewers. We measured the information related to the maximum power point under PSC2 conditions from new in MATLAB software platform. Namely, power (2860W), voltage (313.1V), and current (9.13A). The corresponding data, as well as the calculation part, were carried out related to this is our oversight. None of the five algorithms tracked excessively to the maximum power point. But ISMA still tracks best. We analyzed this phenomenon: 2800 W for the local optimum point (LMPP) and 2880 W for the maximum power point (GMPP), which is a small difference. The algorithms no longer continue to search for an optimum. In order to verify the reasonableness of our idea, we set the radiance to (400/500/600), and the maximum power point is located on the right side under this condition, and the tracking is shown in the figure.At this point, the theoretical maximum power point is 3140 W and the tracked power point of the ISMA algorithm is 3137 W. This shows that the algorithm is still valid in the presence of the global maximum power and the local maximum power. Regardless of whether the maximum power point is located in the middle or right side of the entire P-U curve.(The online submission version does not recognize images, so we hope reviewers understand that we will place your response directly before the manuscript.)

Q4: The declaration of figure 7 has been cited and appeared after the figure itself. It should be before the figure itself.

Thank you. We have moved the text section forward.

Q5: For figures 8,9, and 10, It is better to merge the PV power for the five MPPT at same figure to be easier for the comparison purpose. Also, please do the same suggestion for PV current, voltage and duty cycle.

Thank you to the reviewers. We started to place the comparison curves in a curve box, but found many problems. One is that there are many overlapping curves, which is not conducive to comparison; second, the visual effect is cluttered. We finally chose the expression of this paper and hope the reviewers understand.

Q6:In table 2, It is better to calculate the energy efficiency instead of the power efficiency.

We thank the reviewers for their proposals. The idea of our comparison is that the tracking power of the proposed control algorithm is compared with the maximum power point of the PV generation observed in advance by control and measurement, and the comparison between the two is used as a feedback of the strengths and weaknesses of our control performance. To some extent, it is consistent with the purpose of energy conversion efficiency proposed by the reviewers.

Q7:The authors applied their study under static partial shading condition, PSC1 or PSC2 or PSC3 separately. What is the response of the MPPT techniques if they applied dynamic partial shading conditions? This will discriminate and differentiate between all five MPPT algorithms clearly where some MPPT algorithm succeeded to track the global peak (GP) under PSC1 for example.

Q8: It is better to apply different partial shading conditions with different position (PSC1: GP at the beginning, PSC2: GP at the middle, GP at the end) and values of global peak power.

Q9:The simulation results have no additional contributions, and less value analysis and discussion (Line 377-381). The P-U curves of partial shading conditions under study in the experimental results should be introduced and the experimental results (Theoretical and actual) should be discussed and analyzed deeply.

Thank you to the reviewers. These are a few questions that we have chosen to propose a response to. We are unanimously committed to studying PV power generation systems in the presence of shadows, and the subsequent research directions mainly contain: verification of algorithms for different shading cases, consideration of the photothermal properties of unevenly illuminated PV modules, and we hope to invite reviewers again to review our next work. At present, our experimental platform is being rebuilt and there is no condition for P-U curve, so we hope the reviewers understand. As for the experimental part, we have improved the results in the paper and made a detailed discussion.

Q10:There are some existing studies applied the Slime Mould Algorithm to track the global peak under partial shading conditions like*. Therefore, the authors should review and add all existing studies related to the Slime Mould based MPPT Algorithm versions, provide the research gaps in the previous studies and highlight the improvement and contributions done for the proposed Slime Mould Algorithm.

*Mirza, A. F., Mansoor, M., Zhan, K., & Ling, Q. (2021). High-efficiency swarm intelligent maximum power point tracking control techniques for varying temperature and irradiance. Energy, 228, 120602.

Thank you to the reviewers. We have included the reviewer's proposed paper as a reference for this study. We would like to ask the reviewers to recommend more literature, which will guide our research.

Q11: In abstract, the authors said, “The experimental outcomes show that the shaded PV global maximum power tracking control strategy based on the improved Slime Mould algorithm has superior performance indexes of the tracking process and effectively reduces the power loss in the PV power generation process”, this is not clear and have not been discussed clearly in the experimental results where there are some techniques has similar results to ISMA.

Thanks to the reviewers, we have reconstructed the experiment. In addition, the expression of the abstract section has been improved by changing "1" to "Thanks to the reviewers, we have reconstructed the experiment. In addition, the expression of the abstract section has been improved by changing "The experimental outcomes show that the shaded PV global maximum power tracking control strategy based on the improved Slime Mould algorithm has superior performance indexes of the tracking process and effectively reduces the power loss in the PV power generation process" to "The experimental outcomes show that the shaded PV global maximum power tracking control strategy based on the improved Slime Mould algorithm possesses the best tracking effect with fewer power fluctuations."

Q12:There are some minor errors that should be modified as follow:

We thank the reviewers for their professional attitude. We have revised the details suggested by the reviewers. Thanks again!

Reviewer 3 Report

In this paper a new stochastic optimizer, which is called slime mold algorithm (SMA), based on the oscillation mode of slime mold in nature is applied to the stochastic optimization of the MPPT in partial shading conditions. The paper is properly structured, follows many similar papers proposing advanced MPPT algorithms. The added value of the paper lies in application of the recent algorithm, slight modification and comparison to other similar solutions, including basic laboratory tests. 

Novelty and significance of the paper is limited but it can attract a decent number of citations, because of very actual topic , related to MPPT of photovoltaics.

There are many grammar and editorial mistakes. The paper should be revised and improved.

Author Response

First of all, special thanks to the reviewers for their recognition of our work. We will continue to work on photovoltaic power generation, shading situations, and intelligent algorithms afterwards. Feeling the rigor of the reviewers, we have improved the manuscript part of the paper. Thanks again!

Round 2

Reviewer 1 Report

I really appreciate authors responding to reviewers' comments and making improvements in the manuscript based on comments. However, I am not sure what improvements they made in the manuscript. The authors failed to point out the improvements they made in the response section or the in the manuscript they didn't make any changes they made, so I have no content here to review. So I will be leaving my recommendation as it is without changing it.   

Author Response

We thank the reviewer for the professional attitude. We will send the corrections to the paper in the form of a document.

Reviewer 2 Report

Please, find attached file of the reviewer comments.

Author Response

(The authors gave the same response as above.)

Round 3

Reviewer 1 Report

There are some minor revisions for editing and proofreading is required. There are some formatting issues are there with large white spaces (Page 14). Also, Authors should mark the changes they made on the manuscript with a different color and submit that copy too for the reviewers to get the review done quickly, which they failed to do both times. 

I am recommending it for publication after the editors' proofreading. 

Author Response

Thanks for the reviewer, we have moved the text section at the back to the margin on page 14. In addition, we apologize for not marking each revision. This time we have marked the revisions as highlighted. Thanks again!

Reviewer 2 Report

The authors made many improvements that Ok but there are still minor revisions/modifications required in the attached file. Comments have been provided for some reviewer comments. Please modify accordingly and reply to the comments.

As the authors know, the reviewer comments just to improve the quality of the manuscript. So please do your best to improve the manuscript based on these comments and take these suggestions/comments from the positive side.

Author Response

we are very grateful for the professional attitude of the reviewer, and our result have been enhanced by the reviewer' guidance. We hope that our future results will still be reviewed by you, therefore, we would like to get the reviewer's correspondence email address so that we can recommend you as our reviewer in the future.Thanks again!!!